# The Kinetics of Inflammation-Related Proteins and Cytokines in Children Undergoing CAR-T Cell Therapy—Are They Biomarkers of Therapy-Related Toxicities?

**DOI:** 10.3390/biomedicines12071622

**Published:** 2024-07-21

**Authors:** Paweł Marschollek, Karolina Liszka, Monika Mielcarek-Siedziuk, Iwona Dachowska-Kałwak, Natalia Haze, Anna Panasiuk, Igor Olejnik, Tomasz Jarmoliński, Jowita Frączkiewicz, Zuzanna Gamrot, Anna Radajewska, Iwona Bil-Lula, Krzysztof Kałwak

**Affiliations:** 1Department of Pediatric Bone Marrow Transplantation, Oncology, and Hematology, Wroclaw Medical University, Borowska 213, 50-556 Wroclaw, Poland; liszka.karolina.magdalena@gmail.com (K.L.); mmielcarek@gmail.com (M.M.-S.); idachowska@yahoo.com (I.D.-K.); hazenatalia@gmail.com (N.H.); annapanasiuk@yahoo.pl (A.P.); olejnik@olejnik.x.pl (I.O.); tjarmo@wp.pl (T.J.); jowitafr@gmail.com (J.F.); z.gamrot@gmail.com (Z.G.); 2Division of Clinical Chemistry and Laboratory Hematology, Department of Medical Laboratory Diagnostics, Faculty of Pharmacy, Wroclaw Medical University, Borowska 211A, 50-556 Wroclaw, Poland; anna.radajewska@student.umw.edu.pl (A.R.); iwona.bil-lula@umw.edu.pl (I.B.-L.)

**Keywords:** chimeric antigen receptors, CAR-T cells, CRS, ICANS, cytokines, biomarkers, safety

## Abstract

CD19-targeted CAR-T cell therapy has revolutionized the treatment of relapsed/refractory (r/r) pre-B acute lymphoblastic leukemia (ALL). However, it can be associated with acute toxicities related to immune activation, particularly cytokine release syndrome (CRS) and immune effector cell-associated neurotoxicity syndrome (ICANS). Cytokines released from activated immune cells play a key role in their pathophysiology. This study was a prospective analysis of proinflammatory proteins and cytokines in children treated with tisagenlecleucel. Serial measurements of C-reactive protein, fibrinogen, ferritin, IL-6, IL-8, IL-10, IFNγ, and TNFα were taken before treatment and on consecutive days after infusion. The incidence of CRS was 77.8%, and the incidence of ICANS was 11.1%. No CRS of grade ≥ 3 was observed. All complications occurred within 14 days following infusion. Higher biomarker concentrations were found in children with CRS grade ≥ 2. Their levels were correlated with disease burden and CAR-T cell dose. While cytokine release syndrome was common, most cases were mild, primarily due to low disease burden before lymphodepleting chemotherapy (LDC). ICANS occurred less frequently but exhibited various clinical courses. None of the toxicities were fatal. All of the analyzed biomarkers rose within 14 days after CAR-T infusion, with most reaching their maximum around the third day following the procedure.

## 1. Introduction

CD19-targeted chimeric antigen receptor T-cell (CAR-T) therapy has revolutionized the treatment of relapsed or refractory pre-B acute lymphoblastic leukemia (pB-ALL), improving the prognosis for children in whom conventional therapy has failed [1,2,3]. The latest update from the ELIANA trial [4] revealed impressive outcomes, with a 3-year event-free survival (EFS) of 44% and an overall survival (OS) of 63%. Despite its efficacy, the therapy can be associated with acute toxicities related to immune activation [5,6]. The two most common complications are cytokine release syndrome (CRS) and immune effector cell-associated neurotoxicity syndrome (ICANS). CRS usually presents with fever, hypotension, hypoxia, coagulopathy, and capillary leak, while ICANS includes a diverse set of neurological symptoms, ranging from headache and mild motor disturbances to severe cognitive and motor impairment. Both conditions can be fatal. The prevalence and severity of these toxicities vary between studies [1,2,7]. Significant advances in understanding the pathophysiology, diagnostics, and management have been made over time, resulting in specific drugs and strategies to prevent and treat these complications [8,9,10]. Currently, attention is focused on initiating appropriate procedures early to reduce the risk of severe toxicities, with evidence showing that this approach does not compromise CAR-T cell efficacy [9]. The most important risk factors for severe toxicities include high disease burden, CAR-T cell construct, and a history of intensive treatment [9,11,12,13].

The pathophysiology of these toxicities is mainly based on T-cell activation and proliferation, which trigger macrophage activation and endothelial dysfunction (Figure 1) [14,15,16]. Proinflammatory cytokines released from activated immune cells play a key role in the course of CRS and ICANS, with IL-1, IL-6, IL-8, IFNγ, and GM-CSF being of greatest importance [5,14,17]. Therefore, drugs used for toxicity management, apart from corticosteroids, are targeted towards specific molecules or their receptors. The most significant are tocilizumab (an IL-6 receptor antagonist) [8] and anakinra (an IL-1 receptor antagonist) [15,18]. Limited but promising data are also available on the use of siltuximab (an IL-6 antagonist) [13,19] and emapalumab (an IFNγ-blocking antibody) [20,21,22].

## 2. Materials and Methods

This study is a prospective analysis of selected proinflammatory serum proteins and cytokines in pediatric patients treated with CAR-T cell therapy. It was designed with reference to demographic data, disease burden before therapy, early clinical course after infusion, and particular endpoints (remission, relapse, or death). The study included 27 children who received CAR-T cell therapy between February 2021 and November 2023 at the Department of Pediatric Bone Marrow Transplantation, Oncology, and Hematology of Wroclaw Medical University. After obtaining informed consent from each patient, serial measurements of proinflammatory biomarkers (C-reactive protein [CRP], fibrinogen, ferritin, IL-6, IL-8, IL-10, IFNγ, and TNFα) were taken at specific timepoints: before treatment (day 0) and on days +3, +7, +10, and +14 following CAR-T infusion. Assessments of CRP, ferritin, fibrinogen, and IL-6 were performed in the hospital’s local laboratory on a daily basis. The concentrations of remaining cytokines (IL-8, IL-10, IFNγ, and TNFα) were determined in frozen serum using commercially available ELISA tests: Human IL-8 ELISA (RayBiotech^®^, Peachtree Corners, GA, USA), Human IL-10 ELISA (RayBiotech^®^ Peachtree Corners, GA, USA), Human IFN gamma High-Sensitivity ELISA Kit (Abcam^®^, Cambridge, UK), TNF alpha Human ELISA Kit, Ultrasensitive (Invitrogen^®^, Waltham, MA, USA). All of the measurements were performed in duplicate according to the manufacturer’s manuals. The absorbance was detected at 450 nm (Spark Multimode Microplate Reader, Tecan Trading AG, Zurich, Switzerland). Some samples exceeded the device measuring level when the IFNγ concentration was measured.

The characteristics of the study group are presented in Table 1.

Before CAR-T cell infusion (tisagenlecleucel, Kymriah^®,^, Basel, Switzerland), all patients received standard lymphodepleting chemotherapy (LDC): fludarabine (4 × 30 mg/m^2^ on days −7 to −4 before CAR-T infusion) and cyclophosphamide (2 × 500 mg/m^2^ on days −7 and −6 before CAR-T infusion). The majority of patients had low or undetectable minimal residual disease before LDC. After the infusion, children were observed for at least 14 days for the occurrence of complications, including cytokine release syndrome (CRS) and neurotoxicity (ICANS). The severity of these complications was assessed using the ASTCT Consensus Grading [6]. On days +7 and +14 after the infusion, the CAR-T cell count in the peripheral blood was assessed using a commercially available flow cytometry panel (Miltenyi Biotec, Bergisch Gladbach, Germany). To perform this assessment, CAR-T cells were first specifically bound with a biotinylated CD19 antigen (CD19 CAR Detection Reagent, anti-human, Biotin, REAfinity, Miltenyi Biotec, Bergisch Gladbach, Germany). Subsequently, the labeled CAR-T cells were stained with a fluorochrome-conjugated anti-biotin antibody (anti-biotin, REAfinity, Miltenyi Biotec, Bergisch Gladbach, Germany). The control tube was incubated with the same antibody mix without the Detection Reagent. Finally, the cells were assessed on a CANTO (BD Bioscience, Franklin Lakes, NJ, USA) flow cytometer. Figure 2 presents a representative dot plot used for flow cytometry analysis. The initial step involves identifying cells based on forward and side scatter (FSC and SSC) characteristics. CD45-positive cells were then detected by gating on the CD45 marker. Subsequently, 7-Amino-Actinomycin D (7-AAD) staining was employed to distinguish viable cells, ensuring only 7-AAD-negative (viable) cells were selected. Among these viable cells, CD3-positive cells were identified by staining with a CD3-specific antibody. Finally, within the CD3-positive population, cells expressing the CD19 chimeric antigen receptor (CAR) were detected and quantified using specific fluorescently conjugated antibodies against the CD19 CAR.

The results were interpreted using univariate statistical analysis. Due to the small size of the group and the lack of normal distribution, most of the analyzed quantitative variables are presented as median and interquartile range (Q1–Q3) or range. Nonparametric tests, including the Mann–Whitney U test, Wilcoxon test, and Kruskal–Wallis ANOVA, were used to compare quantitative variables in two or more groups, respectively. Correlations were assessed using Spearman’s rank correlation. If necessary, a logistic regression model was used to confirm the relevance of associations detected in univariate analysis. A *p*-value of <0.05 was considered significant, and values between 0.05 and 0.1 were assessed as indicating a trend toward significance.

The study was conducted according to the principles of the Declaration of Helsinki (version 7, October 2013). Consent from the Institutional Review Board (no. KB-75/2021, dated 8 February 2021) was obtained.

## 3. Results

The median observation time was 12 months. CRS was diagnosed in 77.8% of children, and ICANS was observed in 11.1%. No cases of CRS grade ≥ 3 were observed. Only one child required ICU admission. More data are provided in Table 2. To mitigate toxicities, tocilizumab was administered to 10 patients (37%), mostly at a dosage of 2–4 doses every 8 h until fever resolution. One child received tocilizumab preemptively due to a very high disease burden characterized by 88% leukemic blasts in the bone marrow. Anakinra was used in one patient, and two children received corticosteroids (dexamethasone) to address neurotoxicity. All complications occurred within 14 days following CAR-T cell infusion. During the onset of CRS, we did not diagnose active infection in any patient; however, considering the high risk of systemic infection associated with febrile neutropenia, every patient with fever received antimicrobial treatment in accordance with local center policy.

The concentrations of analyzed biomarkers in the patients’ sera are shown in Table 3. Their distribution and kinetics in the analyzed cohort are presented in Figure 3. In all cases, a statistically significant increase was demonstrated in the days following CAR-T lymphocyte infusion compared to the initial pre-infusion levels (*p* < 0.001).

The higher medians of maximum proinflammatory serum proteins and cytokines concentrations after infusion were found in the group of patients in whom CRS was observed. However, a comparison between no CRS and CRS 1 groups revealed the differences as being not statistically significant. Only the comparison of patients with more severe CRS (≥2) revealed significant changes (Table 4). When comparing baseline biomarker concentrations, patients with CRS grade ≥2 had significantly higher levels of IFNγ before infusion (*p* = 0.03).

We found significant positive correlations between disease burden (defined as MRD value before LDC) and the maximum concentrations of ferritin (R = 0.41, *p* = 0.035) and IL-6 (R = 0.44, *p* = 0.02) following the infusion. A trend toward significance was also observed for IFNγ (*p* = 0.09) and IL-8 (*p* = 0.09). Additionally, there was a significant positive correlation between the number of administered CAR-T cells and the maximum concentrations of CRP (R = 0.44, *p* = 0.02), IFNγ (R = 0.44, *p* = 0.02), and IL-8 (R = 0.54, *p* = 0.003). Significant correlations are presented in Figure 4. There was also a trend toward significance between the amount of infused CAR-T cells and CRS severity (*p* = 0.07). CAR-T cell expansion in peripheral blood (count measured on day +14) was significantly higher in patients with CRS grade ≥ 2 (*p* = 0.03). There was no association between CAR-T expansion and the concentration of analyzed serum molecules.

The univariate analysis revealed that patients who died during the observation period due to disease relapse or progression had significantly higher maximum concentrations of ferritin (2797 ng/L vs. 20,076.6 ng/L, *p* = 0.03), IL-6 (11.5 pg/mL vs. 1084.5 pg/mL, *p* = 0.004), IFNγ (26.4 pg/mL vs. values above test linearity, *p* = 0.004), and IL-8 (52.7 pg/mL vs. 111.91 pg/mL, *p* = 0.03). Multivariate logistic regression, which also considered MRD before LDC, identified IFNγ as the only significant risk factor for death (OR = 1.003, *p* = 0.002).

## 4. Discussion

Toxicity mitigation is crucial to fully exploiting the potential of CAR-T cell therapy. Both preclinical and clinical studies place cytokine release at the center of inflammation-associated complications. Identifying key molecules in the pathophysiology of these toxicities facilitates their potential inhibition and ensures more targeted therapy, especially considering that the impact of glucocorticoid exposure on CAR-T cell efficacy remains unclear [23,24,25]. In our study, the incidence and onset of CRS were almost identical to those reported in the ELIANA trial [1] (77.8% vs. 77.2%, occurring mostly on the third day after infusion). However, in our study, the severity was dramatically lower. It should be noted that this comparison is slightly biased due to the use of different grading scales. The Penn grading scale used in ELIANA qualifies hospitalization for CRS management as grade 2, making it less appropriate for our local setting. Real-world evidence from the CIBMTR [2] revealed a lower CRS incidence, but 16.1% of cases were grade ≥3, whereas we did not observe any cases of grade 3–4 toxicity. The frequency of ICANS was lower compared to the aforementioned studies, but its clinical course was highly varied, including cases of grade 4 neurotoxicity.

As pointed out by various authors [1,2,12], achieving a low tumor burden before therapy significantly decreases the risk of high-grade toxicities. Our aim was to reduce the minimal residual disease as much as possible, preferably to negative levels. This was achieved through individualized selection of bridging therapy, depending on disease burden, clinical status, and previously used treatment lines. We used both conventional chemotherapy (ranging from mild treatments like 6-mercaptopurine or vincristine to more intensive regimens for high-risk patients) and immunotherapy, often combining these approaches. The frequency of administering inotuzumab, ozogamicin, and blinatumomab was notably higher compared to other centers [2,12].

After CAR-T infusion, we observed a significant increase in all analyzed serum biomarker concentrations, both lymphocyte- and monocyte-derived, confirming the activation of effector cells following the procedure. Apart from ferritin and IL-6, other proteins reached their maximum concentration around day +3, which is consistent with the clinical onset of toxicities. None of the analyzed markers appeared to precede the symptoms. Surprisingly, there were no significant differences in serum biomarker levels between patients not presenting with CRS and those with grade 1 CRS. Only more severe cases (grade ≥2) were associated with a statistically significant increase in specific protein concentrations (CRP, ferritin, IL-6, IL-8, and INFγ). This finding is consistent with the study by Teachey et al., which revealed significant cytokine increases in patients with grade 4–5 CRS [14]. This may indicate the potential of these cytokines to predict patients at higher risk of life-threatening toxicities, as the peak appears to slightly precede the greatest severity of symptoms. Grade 1 CRS is common, mostly self-limiting, and can be managed using antipyretics and antimicrobial drugs according to local center policies.

We observed a robust increase in IFNγ shortly after infusion in patients who died during the follow-up period. This was confirmed through multivariate analysis, which identified IFNγ as a predictor of death after CAR-T cell therapy. It is worth emphasizing that none of the deaths were directly caused by immunological complications following the infusion; the children died due to relapse or infection. An interesting case involved a patient with a pre-infusion IFNγ concentration above the level of detection who developed prolonged moderate CRS and very severe ICANS after the CAR-T infusion. Increasing evidence [20,21,22] indicates IFNγ, along with IL-6 and IL-1, as a key cytokine in the pathophysiology of complications after CAR-T cell therapy.

CAR-T cell dose is a known risk factor for severe CRS [26,27]. Our analysis also revealed an association between specific biomarkers (CRP, IL-8, IFNγ) and the number of infused cells. This raises the question of whether increasing the CAR-T dose may elevate the risk of toxicities without improving therapy efficacy. A dose of 1×10⁶ cells per kg seems adequate to ensure a favorable clinical effect [28].

Similar to the research by Hay et al. [26], our study found that patients with more severe CRS had higher CAR-T cell expansion in the peripheral blood. In our study, this difference was not significant on day +7 but became relevant two weeks post-infusion. This aligns with the study by Gust et al. [29], where the highest incidence of complications occurred during rapid CAR-T cell expansion, preceding the peak in peripheral blood. These data suggest that toxicity is primarily mediated by the cytokine surge during CAR-T cell proliferation.

In our study, we observed three cases of neurotoxicity, each with different severity and clinical courses. While this number does not allow for major conclusions, it provides a basis for further analysis. According to the literature [16], the pathophysiology and management of neurotoxicity are complex and insufficiently understood. Diagnostic difficulties may arise because many cytokines are produced directly in the CNS, and some toxicities may not be immunologically derived. Cytokine concentrations can vary between blood and cerebrospinal fluid (CSF) [29], and CSF analysis may be useful for more targeted therapy. Additionally, increasing reports suggest that therapeutic lumbar puncture with intrathecal steroids and methotrexate administration is an efficient way to treat severe neurotoxicity [30,31]. Another important issue is the penetration of drugs through the blood–brain barrier, which limits the effectiveness of treatments like tocilizumab for ICANS [10,16,17,18,32].

Further advancements are necessary in developing agents that can effectively mitigate excessive immune activation following CAR-T cell infusion. Drugs like tocilizumab and anakinra are currently widely utilized and proven highly effective. The approval of emapalumab is deemed crucial given the substantial role of IFNγ in the pathogenesis of toxicities. Additionally, research is focused on incorporating specific switches or suicide genes into CAR-T cells to enable their deactivation or apoptosis in cases where toxicities persist despite conventional treatments [33,34].

## 5. Conclusions

Cytokine release syndrome (CRS) is a common complication following CAR-T cell therapy; however, most cases are mild, primarily due to a low disease burden before lymphodepleting chemotherapy (LDC). Immune effector cell-associated neurotoxicity syndrome (ICANS) occurs less frequently but with varied clinical courses. None of the patients died due to immunological complications. All of the analyzed biomarkers rose within 14 days after CAR-T infusion, with most reaching their maximum around the third day post-treatment. However, significant increases occur in patients with more severe CRS (grade ≥2). Both disease burden and the number of infused CAR-T cells may enhance cytokine release. A high IFNγ surge after infusion may predict an unfavorable outcome, even if not directly associated with immunological complications.

## Figures and Tables

**Figure 1 biomedicines-12-01622-f001:**
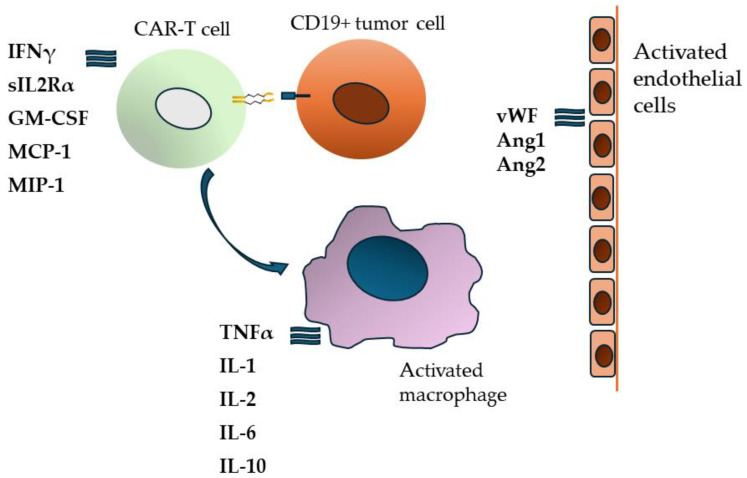
Cytokine release following CAR-T cell activation cascade.

**Figure 2 biomedicines-12-01622-f002:**
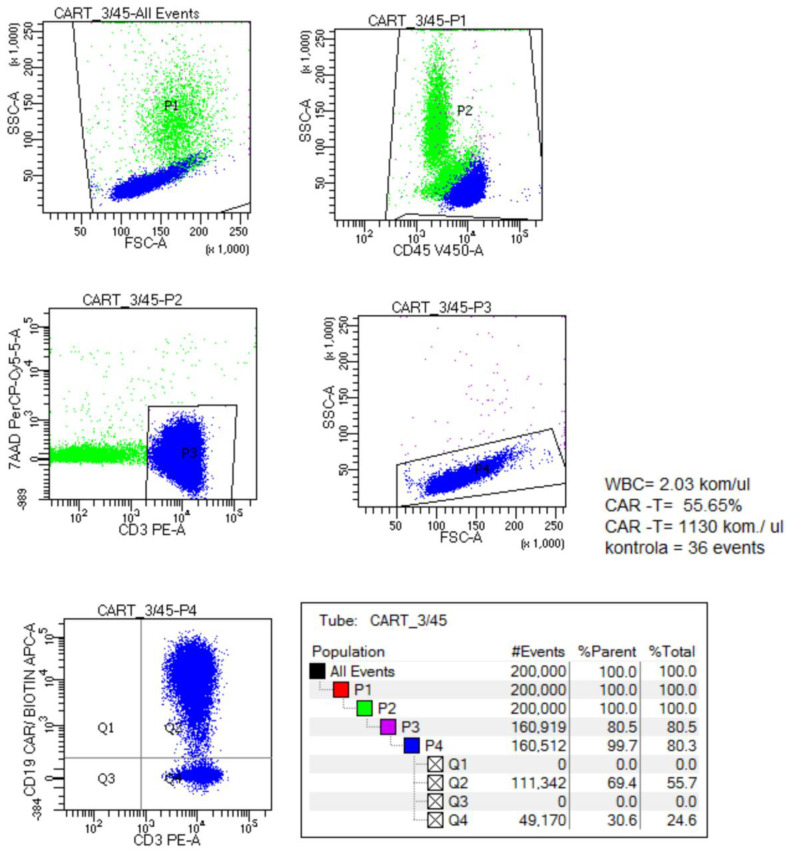
Representative flow cytometry plot for CAR-positive CD3+ lymphocytes detection.

**Figure 3 biomedicines-12-01622-f003:**
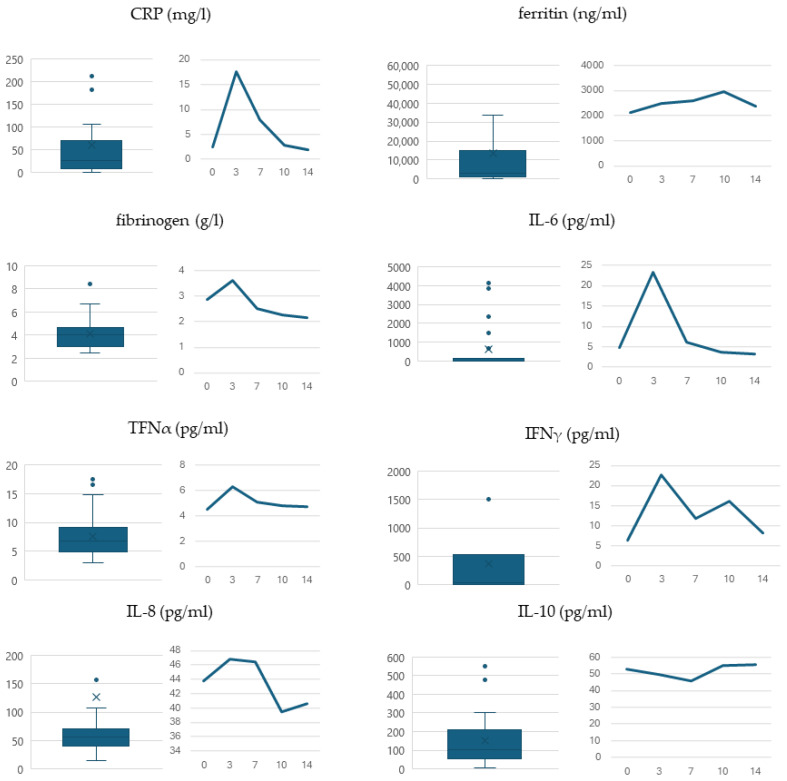
The distribution of maximum biomarker concentrations is shown as boxplots. The kinetics of the analyzed biomarkers over the 14 days following infusion (horizontal axis). The values presented are the medians for each timepoint measurement.

**Figure 4 biomedicines-12-01622-f004:**
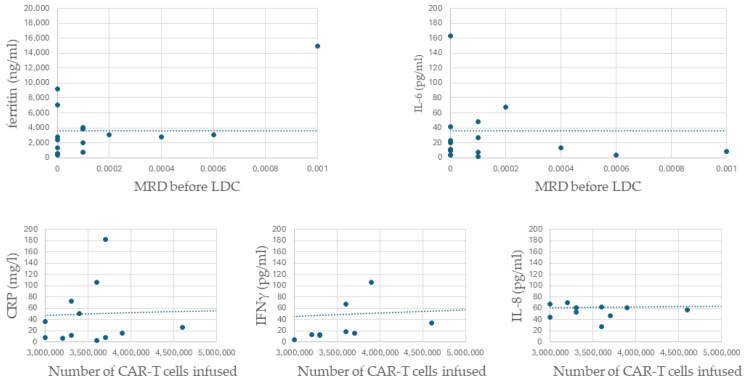
Dot plots for significant correlations. Abbreviations: MRD—minimal residual disease; LDC—lymphodepleting chemotherapy.

**Table 1 biomedicines-12-01622-t001:** Characteristics of the analyzed group.

	N = 27 Patients
Sex	17 boys (62.9%), 10 girls (37.1%)
Age at CAR-T infusion	7.5 years (5.5–12.6 years) ^1^
Indication for CAR-T cell therapy	Relapse after HSCT—12 (44.4%) Secondary refractory disease—7 (25.9%) No eligible HSCT donor—3 (11.1%) Contraindications for HSCT—3 (11.1%) Primary refractory disease—1 (3.7%) Second or later relapse—1 (3.7%)
HSCT before CAR-T	12 patients (44.4%)
Interval between HSCT and CAR-T	10.5 months (8.5–12.5 months) ^1^
Previous treatment with inotuzumab ozogamicin	Before the apheresis—2 patients (7.4%) Bridging therapy—5 patients (18.5%)
Previous treatment with blinatumomab	Before the apheresis—11 patients (40.7%) Bridging therapy—5 patients (18.5%)
MRD before LDC	Negative—9 patients (33.3%) <1 × 10^−4^—4 patients (14.8%) 1 × 10^−4^ < x < 1 × 10^−3^—5 patients (18.5%) 1 × 10^−3^ < x < 1 × 10^−2^—3 patients (11.1%) 1 × 10^−2^ < x < 1 × 10^−1^—4 patients (14.8%) >5%—2 patients (7.4%)
Infused CAR-T cells	median 3.3 × 10^6^/recipient’s kg b.w.

^1^ median and interquartile range (Q1–Q3).

**Table 2 biomedicines-12-01622-t002:** Immunological complications observed in the analyzed group.

Complication	Incidence	Grade	Clinical Course (n; %)
CRS	21 patients (77.8%)	Grade 1—18 patients (85.7%) Grade 2—3 patients (14.3%)	headache—3 (14.3%) hypotension—2 (9.5%) oedema—2 (9.5%) generalized erythroderma—1 (4.8%) hypoxia—1 (4.8%) vomiting—1 (4.8%) fine-wave tremors—1 (4.8%)
ICANS	3 patients (11.1%)	Grade 1—1 patient (33.3%)	Confusion, deterioration of contact
		Grade 2—1 patient (33.3%)	Headache, vertigo, visual impairment, tremor
		Grade 4—1 patient (33.3%)	Deep deterioration of consciousness (GCS 3)

Abbreviations: CRS—cytokine release syndrome; ICANS—immune effector cell-associated neurotoxicity; GCS—Glasgow Coma Scale.

**Table 3 biomedicines-12-01622-t003:** Median concentrations of the analyzed biomarkers at individual timepoints.

Biomarker ^1^	Day 0	Day +3	Day +7	Day +10	Day +14
CRP (mg/L)	2.4 (0.4–392.0)	17.7 (0.5–349.4)	7.9 (0.8–182.0)	2.8 (0.4–50.9)	1.8 (0.3–48.9)
ferritin (ng/mL)	2132.9 (362.3–19,984.4)	2500.4 (289.8–33,511.2)	2600.5 (278.1–144,671.2)	2968.6 (263–67,318.8)	2364.2 (290.1–62,725.6)
fibrinogen (g/L)	2.9 (1.7–8.4)	3.6 (1.9–7.3)	2.5 (1.3–4.3)	2.3 (0.6–3.9)	2.2 (0.8–6.7)
IL-6 (pg/mL)	4.8 (2–598)	23.3 (2–2351)	6.2 (2–4195)	3.6 (2–3829)	3.1 (2–1987)
TNFα (pg/mL)	4.5 (2.2–16.5)	6.3 (3.0–11.3)	5.1 (2.2–14.8)	4.8 (2.5–17.5)	4.7 (2.5–17.1)
IFNγ (pg/mL)	6.4 (0.2–over ^2^)	22.6 (1.6–over ^2^)	11.8 (1.0–over ^2^)	16.1 (0.1–over ^2^)	8.2 (2.3–638.7)
IL-8 (pg/mL)	43.8 (8.9–556.5)	46.8 (15.5–620.2)	46.4 (8.6–over ^2^)	39.4 (9.3–253.3)	40.6 (8.9–349.3)
IL-10 (pg/mL)	52.8 (0.5–365.3)	49.4 (7.3–531.3)	45.9 (1.8–551.4)	55.2 (1.4–479.2)	55.5 (3.4–482.6)

^1^ median and range. ^2^ above the level of determination.

**Table 4 biomedicines-12-01622-t004:** Maximum serum biomarker concentrations in relation to the occurrence of cytokine release syndrome.

Biomarker (Maximum Concentration ^1^)	No CRS	CRS 1	CRS ≥ 2	*p* ^2^
CRP (mg/L)	11.9	26.2	182.0	**0.027**
ferritin (ng/mL)	2938.5	2915.9	15,718.0	**0.03**
fibrinogen (g/L)	3.7	4.1	4.7	0.28
IL-6 (pg/mL)	10.5	21.7	1511.0	**0.028**
TNFα (pg/mL)	5.3	6.6	9.6	0.09
IFNγ (pg/mL)	13.4	34.9	above the level of determination	**0.016**
IL-8 (pg/mL)	52.4	52.8	156.7	**0.034**
IL-10 (pg/mL)	175.2	83.2	134.3	0.85

^1^ median; within 14 days post-infusion; ^2^ comparison between CRS ≥ 2 and rest of the patients; no significant differences between no-CRS and CRS 1 patients’ biomarker levels were observed. Significant *p*-values (<0.05) are marked bold.

## Data Availability

Data are contained within the article.

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
