# Peer review of "The Kinetics of Inflammation-Related Proteins and Cytokines in Children Undergoing CAR-T Cell Therapy—Are They Biomarkers of Therapy-Related Toxicities?"

_biomedicines, 2024, doi:10.3390/biomedicines12071622_

Round 1

Reviewer 1 Report

Comments and Suggestions for Authors

Article by Marschollek et al. concerns a highly pressing problem of acute toxicities accompanying the CAR-T therapy of relapsed or refractory pre-B acute lymphoblastic leukemia (pB-ALL). CD19-targeted chimeric antigen receptor T-cell (CAR-T) therapy is a relatively new genetic engineering – based medical approach, and its side effects and possible complications definitely need further detailed research.

Manuscript is expertly written, undoubtedly the authors possess both deep theoretical and practical knowledge in the field. Conclusions of the paper logically follow from the bulk of the clinical data and raise no objections.

On the manuscript text, I have got two remarks to the authors.

# 1. In many places of the text, you write “proinflammatory cytokines” and list as such a number of proteins under study, namely, C-reactive protein, fibrinogen, ferritin, IL-6, IL-8, IL-10, IFNγ, and TNFα. This occurs in the title, lines 18, 60, 67, 121, 176, 221, etc. (perhaps, I missed some lines). I must say that, terminologically, C-reactive protein, fibrinogen and ferritin clearly are not cytokines, though their functions are related to the inflammation and other pathophysiological conditions. Other mentioned by you proteins are cytokines, indeed. Therefore, I would suggest you to replace the corresponding words in the entire manuscript with more accurate ones, such as “proinflammatory cytokines and other biomarkers”, or “… cytokines and inflammation-related proteins”, or something similar. In the current version, this neglect really spoils the good impression of the manuscript.

#2. Lines 149-150.   “Both preclinical and in vivo studies ….” This is rather incorrect. Majority of preclinical studies are animal experiments, which are in vivo studies, too. I think, you mean here “Both preclinical and clinical studies…”, if I understood your thought correctly.  Or, maybe, you meant here “Both in vitro and in vivo studies…” Please correct this sentence in an appropriate way.

Reviewer 2 Report

Comments and Suggestions for Authors

The work Paweł Marschollek, Karolina Liszka, Monika Mielcarek-Siedziuk, Iwona Dachowska-Kałwak, Natalia Haze1, Anna Panasiuk1, Igor Olejnik, Tomasz Jarmoliński, Jowita Frączkiewicz, Zuzanna Gamrot, Anna Radajewska, Iwona Bil-Lula, Krzysztof Kałwak named “The kinetics of proinflammatory cytokines in children undergoing CAR-T cell therapy – are they biomarkers of therapy-related toxicities?” посвящена важной проблеме минимизации рисков применения CAR-T cell therapy для юных пациентов с relapsed/refractory pre-B acute lymphoblastic leukemia. Acute lymphoblastic leukaemia is predominantly a childhood disease, with approximately 75% of cases occurring in patients younger than 6 years of age. Over the past few decades, therapeutic improvements have improved survival rates for children younger than 15 years of age more than 90%. This has largely been achieved through large cooperative cohort studies with aggressive combinations of conventional chemotherapy including CD19-targeted CAR-T cell therapy. But survival rate of 3 years old patients is 60% and complete victory over the disease has not yet been achieved. A side effect of this therapy is inadequate secretion of pro-inflammatory cytokines, endangering the patient's health and life. Despite the extensive knowledge about the cytokine responses triggered after CAR-T cell therapy, there is no unified approach to minimise side effects.

The authors show that the efficacy of prior treatment steps determines the degree of cytokine release syndrome and immune effector cell-associated neurotoxicity syndrome after CAR-T cell therapy, and the levels of IFNγ secretion after therapy can act as a prognostic marker of outcome. I believe that the paper can be published after a number of additions.

Comments

In the introduction or discussion, it is desirable to include a summary scheme of the triggering of cytokine cascades after CAR-T cell therapy.

Materials and methods. It is desirable to show a typical dot plot for cytometry data and to describe the CAR-T cell gating tactics.

For correlations, it is desirable to include a figure showing the dot plot distribution and trend line, so that the reader can assess the power of the association.

line 150: in vivo should be italic

The specific applied ways of reducing tumour burden compared to other work published in the field should be indicated in the discussion.

In the discussion, it would also be desirable to add a suggestion as to which immune cell inhibitors/activators the authors believe may be promising for reducing excessive cytokine secretion during CD19-targeted CAR T-cell therapies.

 Best regards

Reviewer 3 Report

Comments and Suggestions for Authors

This paper is well written and deals with the well known problem of CRS/ICANS post CAR-T treatment and the role of pro-inflammatory cytokines as predictive factors.

There are unfortunately very few clinical informations reported which make difficult data interpretation: CRS/ICANS treatment is not described, which can have an impact on cytokines levels. The presence of an active infection in the first two weeks post CAR-T is also a frequent finding but here we do not have any information/correlation with cytokines levels.
It is not clear whether cytokines levels were measured in cerebrospinal fluid of patients presenting ICANS.

It is not clear for me if is there any correlation between baseline levels of cytokines and CRS/ICANS and/or CRS/ICANS severity. I understood that cytokine peak correlates with toxicity but not if baseline levels also do it. Also because in lines 178-180 you write : "This may indicate the potential of these cytokines to predict patients at greater risk of life-threatening toxicities" however in your study this seems only to be true for cytokine peak (occurring at the moment of toxicity) and not for baseline levels.

A small issue is also related to tables organisation, particularly table 1 which is not so easy to read - adding lines would probably makes it easier to read it.

Reviewer 4 Report

Comments and Suggestions for Authors

Pawel et al. investigated the cytokine changes induced by CAR-T treatment and their correlation to CRS grade, aiming to identify biomarkers that could potentially predict side effects. While CAR-T cells demonstrate exceptional clinical outcomes, their application is still limited by toxicity, making the identification of these biomarkers crucial for enhancing therapeutic effectiveness. The paper is of significant interest to readers, and the authors have provided sufficient data to support their conclusions. However, there are areas where data presentation could be improved, and further exploration of the results could yield even more valuable information.

1.     Enhanced Data Visualization: Currently, the paper includes tables showing the median or maximum concentration of cytokines. For each cytokine, using bar or dot graphs to display the mean and standard deviation (SD) would provide a more vivid visualization, helping the audience better understand the range of cytokine levels.

2.     Time Course Changes: It would be beneficial to include the time course changes of different cytokines in representative patients. This addition would offer a dynamic view of how cytokine levels fluctuate over time, providing deeper insights into the cytokine response following CAR-T infusion.

3.     Correlation with Tumor Burden and Dosage: Both tumor burden and injection dosage are likely to correlate with cytokine changes. Including graphs that show these correlations would add significant value, helping to illustrate the relationship between these factors and cytokine levels. This additional data could enhance the understanding of how initial disease burden and treatment dosage impact the cytokine response and subsequent toxicity.

By addressing these points, the authors could significantly enhance the clarity and comprehensiveness of their findings, providing readers with a more detailed and informative analysis.

Round 2

Reviewer 1 Report

Comments and Suggestions for Authors

I thank the authors for the thorough work on the manuscript improvement. I have no more questions.

Reviewer 3 Report

Comments and Suggestions for Authors

Thank you for the modifications, I've no further requests

Reviewer 4 Report

Comments and Suggestions for Authors

The reviewer accepts the revised version and has no questions.